# Low-Cost Microbolometer Type Infrared Detectors

**DOI:** 10.3390/mi11090800

**Published:** 2020-08-24

**Authors:** Le Yu, Yaozu Guo, Haoyu Zhu, Mingcheng Luo, Ping Han, Xiaoli Ji

**Affiliations:** School of Electronic Science and Engineering, Nanjing University, Nanjing 210093, China; yule@nju.edu.cn (L.Y.); dg1923016@smail.nju.edu.cn (Y.G.); mf1923107@smail.nju.edu.cn (H.Z.); mingchengluo@163.com (M.L.); hanping@nju.edu.cn (P.H.)

**Keywords:** microbolometer, complementary metal oxide semiconductor (CMOS)-compatible, uncooled infrared detectors, thermal detectors, infrared focal plane array (IRFPA), read-out integrated circuit (ROIC)

## Abstract

The complementary metal oxide semiconductor (CMOS) microbolometer technology provides a low-cost approach for the long-wave infrared (LWIR) imaging applications. The fabrication of the CMOS-compatible microbolometer infrared focal plane arrays (IRFPAs) is based on the combination of the standard CMOS process and simple post-CMOS micro-electro-mechanical system (MEMS) process. With the technological development, the performance of the commercialized CMOS-compatible microbolometers shows only a small gap with that of the mainstream ones. This paper reviews the basics and recent advances of the CMOS-compatible microbolometer IRFPAs in the aspects of the pixel structure, the read-out integrated circuit (ROIC), the focal plane array, and the vacuum packaging.

## 1. Introduction

Infrared (IR) detectors are devices that measure the incident IR radiation by turning it into other easy-to-measure physical phenomenon. The IR detectors may be classified into photon detectors and thermal detectors according to the basis of their operating principle [1]. The photon IR detector absorbs the radiation by the interaction with electrons in the semiconductor material, and then the variation in the electronic energy distribution results in observable electrical output signal. This kind of detectors shows perfect signal-to-noise performance and very fast response, while its utilization is limited because of the requirement of cryogenic cooling [2,3,4,5]. Compared to its competitor, the thermal IR detector, which absorbs the incident IR power to cause temperature rise and measures the consequent change in some physical properties, presents smaller volume, lower cost, and non-necessity of cryogenic cooling, therefore it has wide application in automobile, security, and electric appliance [6,7,8]. The development of thermal IR detectors could be traced back to Langley’s bolometer in 1880, which use two platinum foils to form the arms of a Wheatstone bridge [9]. However, thermal IR detectors failed to attract sufficient attention until the last decade of the 20th century. The reason is that the thermal IR detectors are considered to be much slower and less insensitive than the photon IR detectors [6]. In 1992, both Texas Instruments and Honeywell published their uncooled IRFPAs (infrared focal plane array) based on pyroelectric type and microbolometer type thermal detector, respectively, with fascinating performance [10,11], successfully encouraging a sustained effort to further reduce the pixel size, improve the device performance, and reduce the production cost [12,13,14,15,16,17,18,19,20,21,22,23,24,25,26,27,28,29,30,31,32,33,34,35,36,37,38,39,40,41,42,43,44,45,46].

Today, one of the most attractive thermal IR detectors for imaging purpose is the microbolometer IRFPA. Comparing to other thermal IR detectors like thermopile detector [47,48,49,50], pyroelectric detector [51,52,53,54], and superconducting transition edge sensor (TES) bolometer detector [55,56,57,58], it is promising for the commercial imaging applications because of its respectable performance, small pixel size, and ease to make [59]. Attributing to the continuous efforts and the technological advances, the pixel size of the microbolometer detector fabricated via the low-cost manufacture technology based on silicon LSI (large scale integration) circuit process has been reduced to beyond 17 μm [18,19,20]. Not only does the high-integration process lower the production cost of the detectors, but also it provides mature approach with small feature size and high uniformity to benefit the pixel size and the device performance. Especially, the complementary metal oxide semiconductor (CMOS) microbolometer technology is developed for long-wavelength IR (LWIR, 8–14 μm) FPAs via CMOS foundry compatible approaches [23,24,25,26,27,28,29,30,31,32,33,34,35,36,37,38,39,40,41,42,43,44,45,46]. During the fabrication process, the layer structures of the absorber and the thermal sensor are formed with CMOS process, and then post-CMOS micro-electro-mechanical system (MEMS) process are used to form suspended microbridge structures in purpose of thermal isolation. This technology aims to eliminate the requirement of special process and simplify the post-CMOS MEMS process in order to achieve the ultra-low-cost microbolometer IRFPAs.

However, the most common thermistor materials like vanadium oxide (VO_x_) [60,61,62] and silicon derivatives (a-Si, a-SiGe, a-Ge_x_Si_1−x_O_y_, etc.,) [63,64,65], which have appropriate electrical properties, are not compatible with the CMOS process. For the CMOS-compatible microbolometer IR detector, one choice is the p-n junction diode which has acceptable properties and compatibility with CMOS process; therefore the silicon-on-insulator (SOI) diode IRFPAs have attracted continuous attention since first reported by Ishikawa et al. in 1999 [13], and have been widely adopted in low-cost commercial IR detectors. Besides, CMOS-compatible metal or semiconductor materials (e.g., aluminum [41,42,43], titanium [12,29], polycrystalline silicon [44], etc.,) have been investigated as another choice as well. Although the SOI diode IRFPA and the CMOS-compatible material microbolometer IRFPA have relative low temperature coefficient, it could be compensated by the high integration and high uniformity. Till now, a lot of efforts have been done to improve these two types of microbolometer detectors: Ueno et al. proposed a multi-level structure that has an independent metal reflector between the absorber and the thermistor for interference IR absorption in SOI diode IRFPA [15]; Takamuro et al. invented the 2-in-1 SOI diode pixel technology to significantly increase the diode series number in a pixel, leading to the increase of responsivity [18]; Ning et al. implemented a double-sacrificial-layer aluminum microbolometer fabrication process to enhance both the thermal isolation of the suspended microbridge structure and the IR absorption of the optical resonant cavity [42].

In this paper, we focus on the CMOS-compatible microbolometer IR detectors, that is, the low-cost microbolometer type IR detectors for imaging purpose fabricated via CMOS process (or conventional silicon LSI circuit process). During the fabrication process, no special delicate approach (e.g., the deposition of vanadium oxides) should be needed, and only simple MEMS process is applied after the CMOS process. The basics and the fabrication processes of such low-cost microbolometer IR detectors will be introduced, while the development trends and the technological advances are also discussed.

## 2. Theory and Development Trends

### 2.1. Basics of Microbolometer

When the IR radiation falls on the surface of the bolometer, it is absorbed and results in a temperature increase Δ*T*. When the heat balance is reached, the temperature rise is
(1)$$\Delta T=\frac{\varepsilon P_{0}}{{\left(G^{2}+\omega^{2}C^{2}\right)}^{1/2}}=\frac{\varepsilon P_{0}}{G{\left(1+\omega^{2}\tau^{2}\right)}^{1/2}}$$
here *C* is the thermal capacitance of the absorber, which is connected to the environment via the thermal conductance *G*. *ε* is the emissivity (absorptance) of the incident IR radiation with amplitude *P*_0_ and angular frequency *ω*. *τ* is the thermal time constant, which commonly ranges from several to several tens of milliseconds for the thermal IR detector. For both resistance type and diode type microbolometers, the temperature increase is transferred into the electric signal and then measured.

Lower thermal conductance results in larger temperature increase and higher sensitivity, but worse time constant. Therefore, a small thermal capacitance is always necessary in order to relax the restriction of the trade-off between the sensitivity and the thermal constant time.

The output signal of the microbolometer accompanies with noise that originates from various uncorrelated source, resulting in undesired random fluctuations. There are several major noise sources that should be considered in a microbolometer IR detector: Johnson noise, temperature fluctuation noise, and 1/*f* noise [66]. Besides, the shot noise also could be taken into consideration for the diode type microbolometer detector [27]. The total noise could be calculated in terms of its mean square as the sum of the mean squares of these noises:(2)$$\bar{V_{n}^{2}}=\bar{V_{J}^{2}}+\bar{V_{TF}^{2}}+\bar{V_{1/f}^{2}}+\bar{V_{shot}^{2}}$$

These noises determine the noise equivalent temperature difference (NETD). NETD is defined as the change in temperature when the output signal equals to the noise, i.e., the minimum temperature difference that could be measured. The performance of a microbolometer IR detector with optics may be evaluated in terms of the NETD. It is given by [67].
(3)$$NETD=\frac{\left(4F^{2}+1\right)\sqrt{\bar{V_{n}^{2}}}}{A\varepsilon R_{v}{\left(dP/dT_{t}\right)}_{\lambda_{1}-\lambda_{2}}}$$

Here *F = f*/*D* is the *F*-number of the optical system, where *f* and *D* are the focal length and the aperture of the optics, respectively. *A* is the size of the absorber, *R_v_* is the responsivity defined as the change of the output voltage resulted from per unit incident IR power, (*dP*/*dT_t_*)*_λ_*_1−*λ*2_ is the change in power per unit area radiated by a blackbody at temperature *T_t_* measured within the IR spectral band from *λ*_1_ to *λ*_2_. The value of (*dP*/*dT_t_*)*_λ_*_1−*λ*2_ for a 295 K blackbody within the 8–14 μm band is 2.62 × 10^−4^ W/cm^2^K [68]. The NETD of a low-cost microbolometer IRFPA under its operation condition typically ranges from 50 to 500 mK.

### 2.2. Development Trends

Before the thermal detector has been demonstrated to be practical for imaging purpose, the IR detector field was dominated by the photon detectors which were restricted to military applications because of the expensive materials and requirement of cryogenic coolers. The appearance of the commercialized thermal IR detector encourages the expectation for non-military application. The CMOS-compatible microbolometer IRFPA, from its very beginning, aims to further lower the cost and the chip size, while maintain an acceptable performance.

Pixel size, as the indicator of the integration level, is the key factor limiting the chip size. The pixel size reduction of the CMOS-compatible IR detectors is shown in Figure 1. Because of the considerable efforts, the pixel size of the SOI diode uncooled IRFPAs has been reduced to 15 μm in 2011 [18]. Meanwhile, the spatial resolution or the array size is related to the pixel size. With the progress of the CMOS microbolometer technology, the array size also increases from 128 × 128 reported in 1996 [12] to 2000 × 1000 reported in 2012 [19].

As shown in Figure 2, unlike the pixel size and spatial resolution, the NETD generally shows a trend of remaining in the same level rather than continuously improving. Since the CMOS microbolometer IR detectors mainly aim the non-military market, a NETD of several tens mK is already capable in handling those applications.

Smaller pixels collect less IR power to increase the temperature, resulting in lower sensitivity. In a conventional structure of the microbolometer pixel, the absorber, the thermal sensor, and the supporting legs are in the same suspended layer. When the pixels are scaled down, higher fill factor and higher emissivity are necessary in maintaining the same sensitivity since less IR radiation is absorbed, bringing a hard task for the trade-off between the thermal conductance and the fill factor. To fix this issue, the multi-level structures with hidden support leg [69,70,71] or umbrella absorber [15,72,73] have been proposed. By these new structures, the high fill factor and low thermal conductance could be simultaneously achieved in small pixels. However, it seems that the further step of the pixel size reduction has slowed down in recent year, indicating the requirement of novel technical innovation. In addition, the restriction of the diffraction limitation also impedes the progress of pixel size reduction, which is discussed later.

## 3. Complementary Metal Oxide Semiconductor (CMOS)-Compatible Microbolometer Pixel

### 3.1. The Resistance Type Microbolometer Pixel

Figure 3 shows the pixel structure of the resistance type microbolometer. The microbolometer pixel contains three parts: the infrared absorber, the thermal sensor, and the microbridge structure. The infrared absorber usually consists of the dielectric layer or the multi-layer structure of dielectric and metal layers [74]. The thermal sensor is implemented using a CMOS-compatible thermistor layer sandwiched in the absorber, which is designed to be serpentine to maximize the resistance. The microbridge structure consists of two support legs to sustain the suspended area, creating a thermally isolated cavity between the absorber and the substrate in order to greatly reduce the thermal conductance. In an Al microbolometer, the IR absorber is implemented using the SiO_2_/Si_3_N_4_ layer, with the Al thermistor from the metal interconnect layer Metal 3 sandwiched inside the SiO_2_ layer. The SiO_2_ and Si_3_N_4_ also provide protection for the thermistor and the read-out circuit during the post-CMOS etching process.

As shown in Figure 4, the process flow of the Al microbolometer shown in Figure 3 is as follows:The p+/n−well (2,3), gate oxide (4), and polysilicon (5) are fabricated on the substrate (1) via lithography, deposition, ion implantation, and annealing in order to form the transistor.Deposit SiO_2_ (6) as the isolation layer, then etch and deposit W (7) to form the contacts. Afterwards the metal interconnect layer Metal 1 (and the subsequent metal interconnection layers in the active region as well) is formed by depositing Al (8) as the connection of the read-out circuit.Deposit SiO_2_ (6) and then form the W (7) vias. The Al (8) in metal interconnect layer Metal 2 is deposited as the sacrificial layer in the sensor region.Deposit SiO_2_ (6), form the W (7) vias, and then deposit Al (8) for the interconnect layer Metal 3 to form the thermistor in the sensor region.Deposit SiO_2_/Si_3_N_4_ (6,9) to protect the device. Then dry etch the SiO_2_/Si_3_N_4_ over the pad area and expose the sacrificial layer.Use photoresist (10) to protect the pad area during the post-CMOS etching. Use the phosphoric acid solution to etch the sacrificial layer to form the cavity and expose the microbridge structure.

The steps a to e are in a standard CMOS process, while step f is in a post-CMOS MEMS process. The whole process could be completed in a CMOS foundry to achieve high uniformity devices in ultra-low production cost. However, an intrinsic limitation of the CMOS-compatible microbolometer is the thermistor material. When infrared radiation illuminates the surface of the absorber, the thermistor in the absorber is heated and causes a change in its resistance related to its temperature coefficient of resistance (TCR) *α*, defined as:(4)$$\alpha =\frac{1}{R_{0}}\cdot \frac{dR_{b}}{dT}$$

*R*_0_ is the resistance of the bolometer at room temperature; *dR_b_* is the resistance change depending on the temperature change *dT*. Under a certain bias current, the change of the thermistor resistance could be obtained by measuring the output voltage. Therefore, the value of TCR significantly influences the device sensitivity. Generally, the semiconductor-based microbolometers have negative TCR values, while the metal ones have positive TCR values. Table 1 lists several common CMOS-compatible thermistor materials. Compared to the high TCR thermistor materials like VO_x_ which has a TCR of about 2–3%/K, the CMOS-compatible materials have obvious disadvantage in the TCR. This results in a low sensitivity which needs to be compensated by the high-spec read-out circuit.

### 3.2. The Diode Type Microbolometer Pixel

The pixel structure of the diode type microbolometer is similar to that of the resistance type microbolometer; it also consists of three parts: the infrared absorber, the thermal sensor, and the microbridge structure. Here the thermal sensor becomes the p-n junction diodes, which are connected in series to enlarge the output signal. The diodes are usually fabricated on the SOI film for several reasons: (a) The diodes fabricated on deposited Si film exhibit large 1/*f* noise [76,77]; (b) the diodes fabricated on Si substrate need a special electrochemical etch-stop technique to protect the n−well during the post-CMOS etching process [27,33,78]; (c) the SOI film is expected to have fewer defects and localized states which could reduce the 1/*f* noise. The pixel structure of a SOI diode detector is shown in Figure 5. The BOX (buried oxide) layer and the dielectric film over the diodes protect the diodes during the post-CMOS etching process.

The temperature change in the diode under a certain bias current results in a voltage shift. The temperature coefficient in a diode type microbolometer is determined by the forward voltage *V_f_*. With diodes in series connection, the sensitivity is given by [14]:(5)$$\frac{dV_{f}}{dT}=-\frac{n(1.21-V_{f}/n)}{T}$$
where *n* is the number of the diodes in the series. The typical value of the sensitivity for a single diode at 300 K is ~2 mV/K under a bias voltage of 0.6 V [59], which is equivalent to a temperature coefficient of only ~0.33%/K. However, as the number of the diodes in the series increases, the temperature coefficient could become comparable to the TCR of VO_x_. For instance, when *n* = 8, the diodes in series connection have a temperature coefficient of ~3%/K. Meanwhile, benefiting from the high uniformity of the CMOS process and the low defect density in the SOI film, the diode type microbolometer usually exhibits much better noise.

### 3.3. Improvement in Absorber for Small Pixel Structure

The small pixels benefit the detectors from a production point of view. For instance, when the scaling down from 25 μm pixel to 17 μm, it decrease the processing cost by 40% and the power consumption by 33%, while the detection range is increased significantly [61]. However, since the IR absorption is proportional to the absorber area, it demands novel structures that achieve high fill factor or high emissivity in order to compensate the disadvantage of small pixel size.

The umbrella absorber is a widely adopted design to maximize the absorber area that captures more incident IR energy. As shown in Figure 6a, it consists of an IR absorber layer, which individually suspends over the bolometer and support legs, supported by one or more posts. The umbrella absorber consists of dielectric layer or multi-layer structure of metal and dielectric layers, which is the same as that of the conventional absorber layers. Some umbrella absorbers have etch holes designed to enhance the sacrificial removal. These etch holes also benefit the responsivity due to the decrease of thermal capacitance of the umbrella absorber [79]. The umbrella absorber can achieve a fill factor above 90% and ~23% improvement in responsivity [72]. This structure provides the fill factor close to an ideal value at the expense of more process steps, usually increasing 2–5 masking layers and corresponding deposition and etching steps [45].

Another prospective approach to improve the absorption is the absorber with a metasurface. The magnetic resonance in the metasurface could control the thermal emission of phonon, therefore the IR absorption spectrum of the metasurface could be manipulated via changing the structure parameter [80]. This could be implemented to the surface of the absorber in order to enhance the IR absorption in the microbolometer pixel, as shown in Figure 6b. This novel approach has attracted the attention of several groups and the preliminary results reveal its potential of frequency selection and absorption enhancement [81,82,83].

## 4. Read-Out Integrated Circuit (ROIC)

The IR energy absorbed by the microbolometer pixel is transformed into weak photocurrent, which is not capable for direct processing due to the noise interference. The photocurrent needs to be amplified and finally turned into digital signal by the read-out integrated circuit (ROIC). Benefiting from the CMOS technology, the ROIC has the advantages of high signal handling capacity, high circuit density, low power dissipation, high uniformity and low noise [3]. As shown in Figure 7, the ROIC usually contains several blocks: (1) The read-out circuit (ROC) to amplify the photocurrent and turn it into a voltage signal; (2) the row decoder and the column multiplexer to select an individual pixel; (3) the power supply and clock signal generator to provide the bias and the clock signal; (4) some IRFPAs have the on-chip analog-to-digital converter (ADC) integrated in the ROIC, while others implement the external ADC. Among all these blocks, the ROC and the ADC are the core blocks which determine the performance of the ROIC.

### 4.1. Read-Out Circuit (ROC)

In the ROC, the photocurrent generated from the pixel is amplified and accumulated by a capacitor during an integration time to form a stronger voltage signal, which is then read out into a sample-and-hold (S/H) circuit for the consequent digital conversion in ADC. The design of the ROC significantly affects the power dissipation and the quality of the analog output signal before converting. The most commonly used ROC configuration in microbolometer IRFPAs are direct injection (DI) [61,84,85], gate modulation input (GMI) [13,34,35], and capacitive transimpedance amplifier (CTIA) [20,31,65,86,87]. The design concepts involve the performance and the structural complexity; each designer may prefer a different design depending on the technical requirement and the process schedule.

The structure of the DI configuration is shown in Figure 8a. The photocurrent is injected to C1 to integrate after being amplified via M1, and then is read out to S/H circuit through M4. The function of M2 is to reset the voltage on C1. The DI benefits from simple structure and low power dissipation, but suffers from unstable bias voltage, poor linearity, and poor noise suppression. Figure 8b shows the structure of the GMI configuration. The photocurrent flows into a current-mirror to generate the mirror current toward C1 and then gets integrated. The GMI itself has a varying current gain depending on the background, therefore leading to the higher sensitivity, background suppression, and high dynamic range. Meanwhile, the circuit noise is suppressed by the current mirror structure. The disadvantage of GMI is that the linearity is still affected by the unstable bias voltage, while the current gain and injection efficiency are susceptible to the threshold voltage and process condition of the metal-oxide-semiconductor field-effect transistor (MOSFET), resulting to a negative influence on the circuit performance.

As the most popular configuration in microbolometer IRFPAs, the CTIA configuration is shown in Figure 8c, which is an integrator with the capacitor C1 is in the negative feedback loop of the operational amplifier. M1 is the reset switch and M2 controls the output. The CTIA has low input impedance thus high injection efficiency, stable bias thus excellent linearity, controllable current gain, high sensitivity, and good jam-proof. However, it has relatively high power dissipation, large occupied area, and would introduce more noise due to the offset voltage. Compared to the DI configuration, the CTIA has higher current gain which provides higher sensitivity to detect weaker current, and it also has lower input impedance leading to higher injection efficiency. Compared to the GMI configuration, the CTIA provides more stable bias voltage for the detector, resulting in a better linearity in the output signal. The typical CTIA parameters for microbolometer IRFPAs are shown in Table 2.

### 4.2. Analog-to-Digital Convertor (ADC)

Generally, the high-speed ADC with a high dynamic range is required for the utility in the CMOS microbolometer IRFPAs. Although the on-chip ADCs using the pixel-level Sigma-Delta (Σ-Δ) ADC [92,93,94], the monolithic pipeline ADC [95,96], and the column-parallel successive approximation register (SAR) ADC [97] are reported to be available to achieve the high sensitivity ROICs for microbolometer IRFPAs, there are no report about the on-chip ADC for readily available CMOS microbolometer IRFPAs. The CMOS microbolometer IRFPAs usually use external ADCs, due to the inadequate signal processing area in the monolithic FPA. The microbolometer IRFPAs raises the requirement to the ADC such as low power dissipation, high speed, low delay, low offset voltage, low noise, and high slew rate. Table 3 shows typical parameters of an on-chip monolithic pipeline ADC for the microbolometer IRFPA.

## 5. Focal Plane Array (FPA)

Microbolometer pixels are usually fabricated on the substrate with repeating arrangement to form a microbolometer array for imaging purpose. Each microbolometer absorbs the incident IR radiation and transforms it into electric output, which is read out and calibrated by the ROIC to produce a pixel in a two-dimensional image. A microbolometer FPA is the combination of the microbolometer array and the ROIC. Generally, the IRFPA could be sorted as hybrid and monolithic [98]. In the hybrid FPA, the detector pixels and the ROIC are fabricated in different substrates, which are combined using the flip-chip bonding via metal bumps. Since it has the advantages such as the independent optimization of detector material and multiplexer, near 100% fill factor, and sufficient signal processing area, it is widely used in the cooled IRFPAs and high-end uncooled IRFPAs [6]. The monolithic FPA integrates the ROIC and the detector pixels in the same substrate, and part of the column or row selecting circuit is integrated in the pixels. Since the silicon-based monolithic FPA technology is compatible with CMOS process, providing a mature approach with high uniformity and low cost, it is widely used in the microbolometer IRFPAs.

The reduction of pixel size makes challenging tasks for the mechanical stability of the pixel structure, the ROIC, the signal to noise ratio, etc. Not only the thermal sensor material, but also the overall process becomes the limits of the final performance of the IRFPAs. Table 4 lists the performance of several commercial IRFPAs. The performance of SOI diode IRFPAs and the CMOS-compatible resistance microbolometer IRFPAs is still inferior to that of the VO_x_ or Si derivatives microbolometer, but the gap between the two is small. This means the low sensitivity resulted from the low TCR of the thermal sensor material could be partly compensated by the small feature size and high uniformity provided through CMOS or Si LSI process.

## 6. Vacuum Packaging Technology

The thermal conduction via the atmosphere takes over a large fraction in the total thermal conduction, especially when the pixel size is small. Since the temperature change and thus the responsivity is proportional to the thermal conductance, the vacuum packaging of the microbolometer pixels is necessary to eliminate the thermal conduction through air. Unfortunately, the cost of the vacuum packaging is one of the major cost drivers for the microbolometer IRFPA. The typical vacuum level required here is below 1 Pa, which raises a challenge to the packaging technology [100]. Although such requirement could be achieved via one-by-one pumping through a fine-bore tube, the cost becomes a bottleneck in lowering the cost of uncooled IRFPAs. Figure 9 shows the concept of the wafer-level packaging (WLP) technology for IRFPA, which is a popular option for cost reduction [59,101,102]. In this technology an IR transparent cap wafer is bonded to the IRFPA wafer under vacuum and then the hermetical sealing is achieved using solders. Several steps are needed prior to the bonding to accomplish the cap wafer. The cavities for the pixels are formed via etching, and then both sides of the cap wafer are antireflection-coated, afterwards the vacuum getters are deposited inside the cavities. The WLP technology is a practical technology that is capable to reach an average seal yield > 95% with correct parameters [103].

Although the wafer level packaging technology provides a significant cost reduction, it still takes a considerable proportion in the total cost of the uncooled IRFPA, especially for the low-end market. A pixel level packaging (PLP) technology has been developed to address this issue [104,105,106]. The PLP process consists in the manufacturing of IR transparent microcaps that cover each pixel in the direct consequent step of the wafer level bolometer fabrication, i.e., no extra bonding process is needed. Figure 10 shows the schematics of a packaged pixel. To form this structure, first, a sacrificial layer with trenches around each pixel is formed above the microbolometer via deposition and etching. Then, an IR transparent material is deposited to form the microcap structure. After that, etch holes are formed through the IR transparent microcap and the sacrificial layer is removed. Finally, a sealing and anti-reflecting layer is deposited under high vacuum. The pixel using PLP keeps a stable vacuum level below 10^−3^ mbar and shows nominal performance after one year of ageing, demonstrating the PLP to be a prospective novel vacuum packaging technology for the microbolometer IRFPAs.

## 7. Limitation and Future Trends

The minimum resolvable size *x* decided by the diffraction limitation could be expressed by the *F*-number and the wavelength *λ* according to the Rayleigh Criterion, which is
(6)x≈fθ=1.22λF

Here *θ* is the diffraction angle, and *f* is the focal length of the optical lens. In a LWIR detector the *λ* ranges from 8–14 μm, while the *F*-number for CMOS microbolometer IRFPAs is usually close to 1 to make the device compact, indicating the minimum resolvable size is 10–17 μm. When the pixel size is between 0.5*λF* and 1.22*λF*, the resolution still benefits from the oversampling but saturates quickly as the pixel size is smaller [79]. However, unlike the photon detectors which prefer a pixel size close to or even smaller than the diffraction limit to achieve the maximum performance, the reported CMOS microbolometer IRFPAs are still in the “detector limit” regime, i.e., still far from the potential limiting performance.

The main factor that limits the pixel size reduction in CMOS microbolometer is the responsivity. As mentioned above, smaller pixel means less IR absorption, resulting in low responsivity. Meanwhile, the scale-down of circuits results in a lower applied bias voltage, which also means lower responsivity. The responsivity could be enhanced by adjusting the fill factor, the emissivity *ε*, the thermal conductance *G*, and the temperature coefficient TCR or *dV_f_*/*dT*. The fill factor and the emissivity in the state-of-the-art technology are already high, although the *ε* is still capable to increase to a certain extent via the metasurface technology. The thermal conductance could be decreased with thinner or longer support legs. The TCR is intrinsic to the material, but the resistance increase of the thermistor is able to raise the responsivity. On the other hand, the temperature coefficient of the diode type microbolometer is mainly determined by the number of diodes in series. In any case, the way to enhance the responsivity of the small pixels is related to a smaller feature size.

Besides, the spatial resolution is also affected by the array size. Although the XGA (extended graphics array) format (1024 × 768) is popularized in the VO_x_ and silicon derivatives microbolometer IRFPAs, the QVGA (quarter video graphics array) format (320 × 240) is still popular with the CMOS microbolometer IRFPAs. Since the difficulty to achieve larger array size is much easier compared to that to the pixel size reduction, the status of the low spatial resolution could be considered as a trade-off between the production cost and the performance. It also implies that the market demand to the performance improvement in low-end IR detector is not eager. However, the merit of the pixel size reduction is significant. The small pixel provides low production cost, high spatial resolution, and small device size. Although the steps of the pixel size reduction in the CMOS microbolometer IRFPAs has slowed down in recent years because of insufficient market demand, the smaller pixels with lower costs and better performance will come sooner or later as the technology based on smaller feature size becomes practical.

## Figures and Tables

**Figure 1 micromachines-11-00800-f001:**
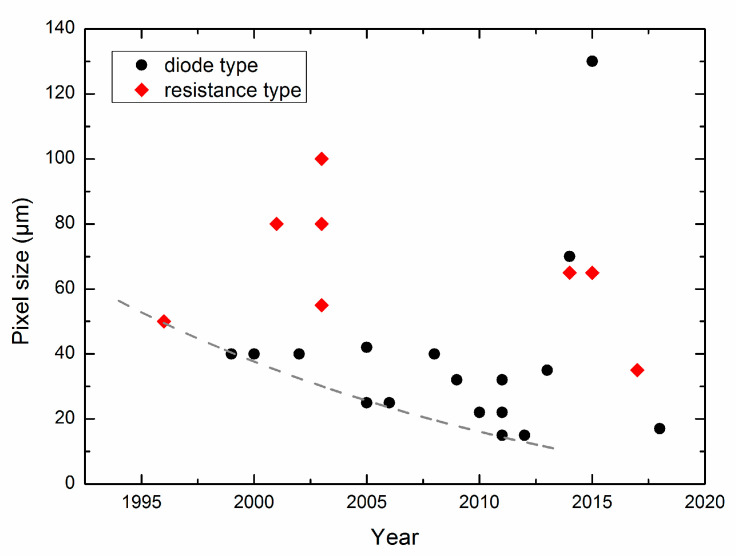
Trends of the pixel size reduction for complementary metal oxide semiconductor (CMOS)-compatible microbolometer infrared (IR) detectors, data taken from (in left-to-right order) [12,13,14,15,17,18,19,20,21,22,25,27,28,29,31,32,33,34,35,37,39,40,41,43,46].

**Figure 2 micromachines-11-00800-f002:**
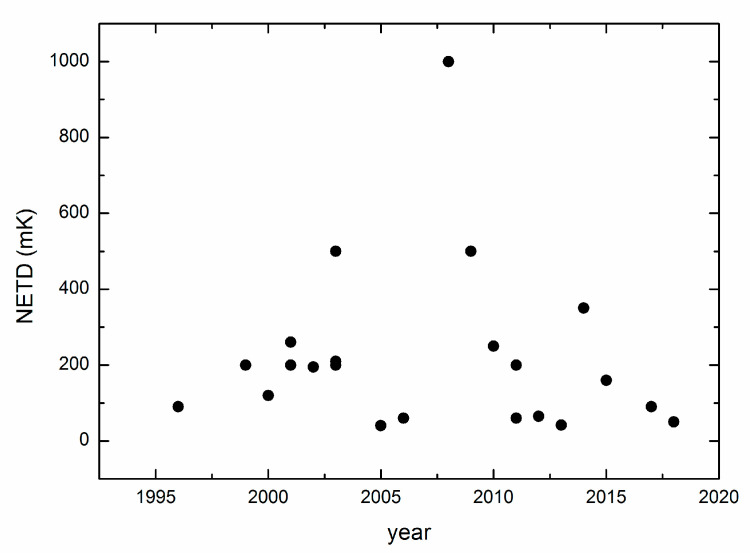
Trends of the noise equivalent temperature difference (NETD) for CMOS-compatible microbolometer IR detectors, data taken from (in left-to-right order) [12,13,14,15,17,18,19,20,21,22,25,26,27,28,29,31,33,34,35,37,40,46].

**Figure 3 micromachines-11-00800-f003:**
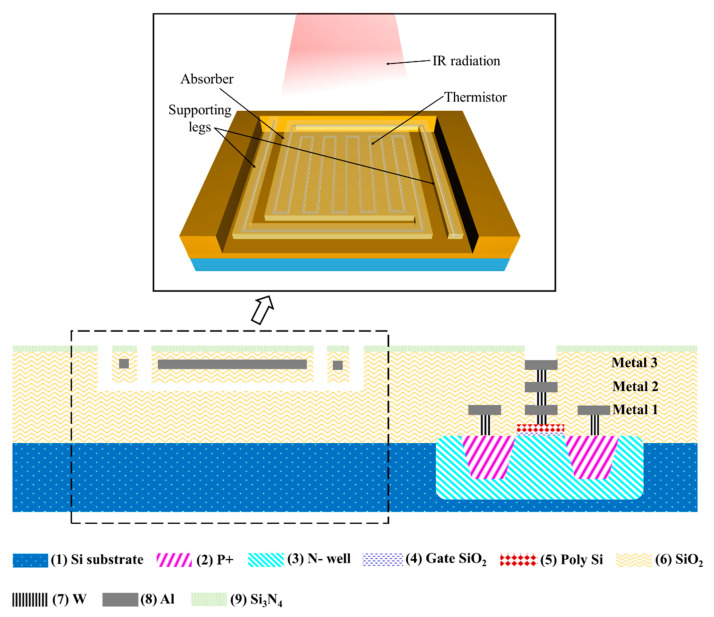
The cross-sectional schematics showing the pixel structure of an aluminum microbolometer. The inset provides a 3D perspective view on the structure of the absorber and the thermistor.

**Figure 4 micromachines-11-00800-f004:**
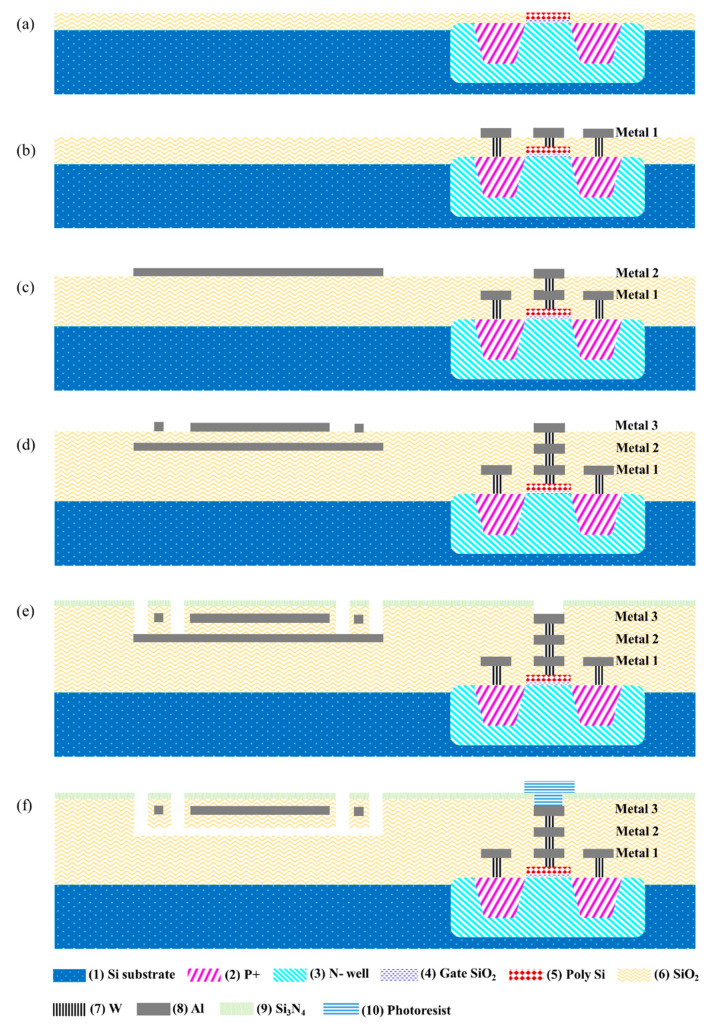
The process flow for an Al microbolometer: (**a**–**e**) are in a standard CMOS process and (**f**) is in a subtractive micro-electro-mechanical system (MEMS) process.

**Figure 5 micromachines-11-00800-f005:**
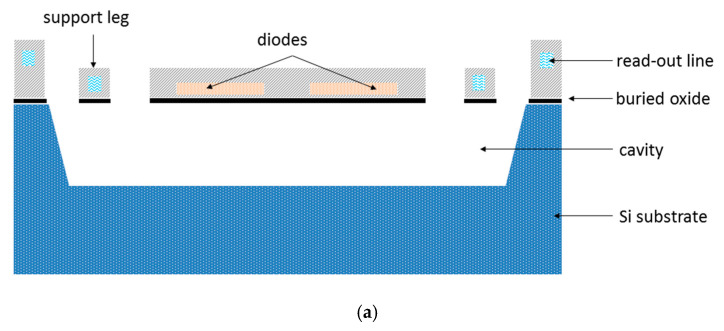
The pixel structure of a diode type microbolometer: (**a**) cross-sectional view and (**b**) 3D perspective view.

**Figure 6 micromachines-11-00800-f006:**
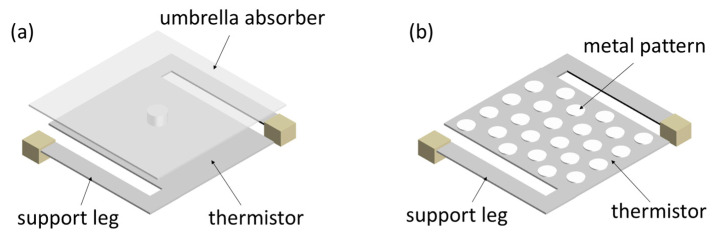
The schematics of a microbolometer with (**a**) umbrella absorber; (**b**) metasurface.

**Figure 7 micromachines-11-00800-f007:**
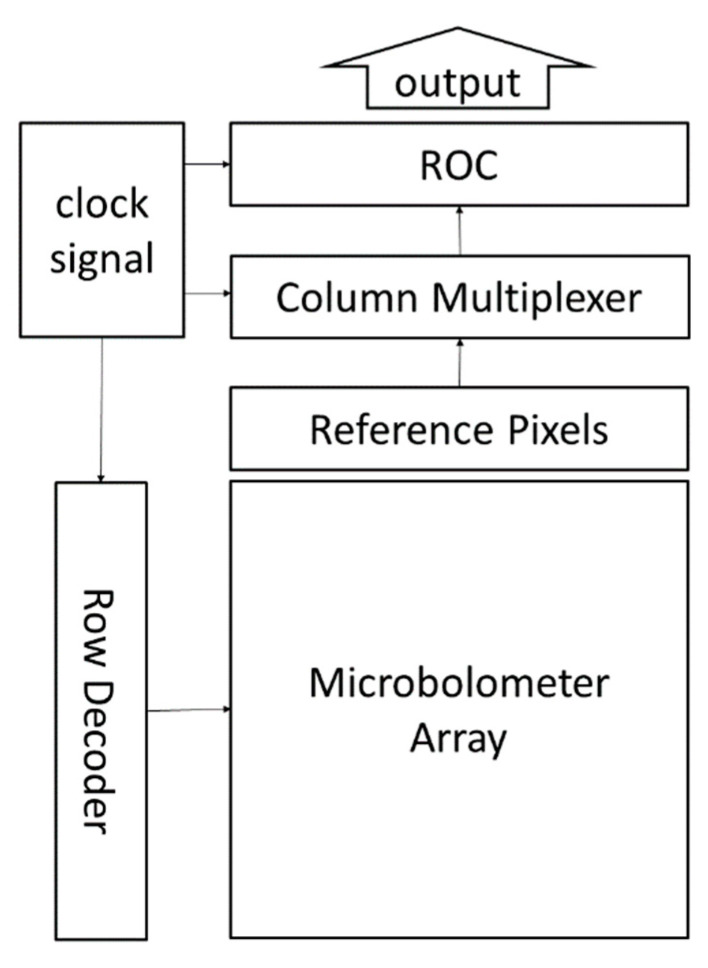
The block schematics of a read-out integrated circuit (ROIC).

**Figure 8 micromachines-11-00800-f008:**
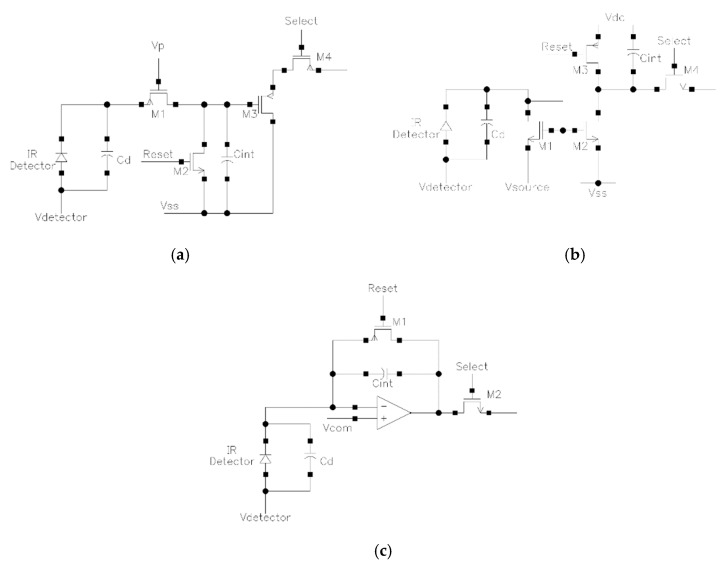
Structures of read-out circuit configuration: (**a**) direct injection (DI), (**b**) gate modulation input (GMI), and (**c**) capacitive transimpedance amplifier (CTIA).

**Figure 9 micromachines-11-00800-f009:**
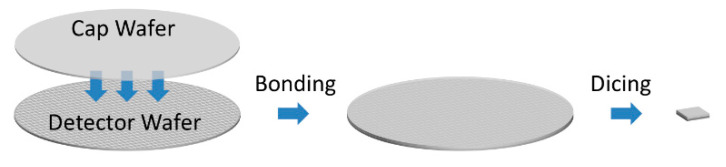
Concept of the wafer level packaging technology.

**Figure 10 micromachines-11-00800-f010:**
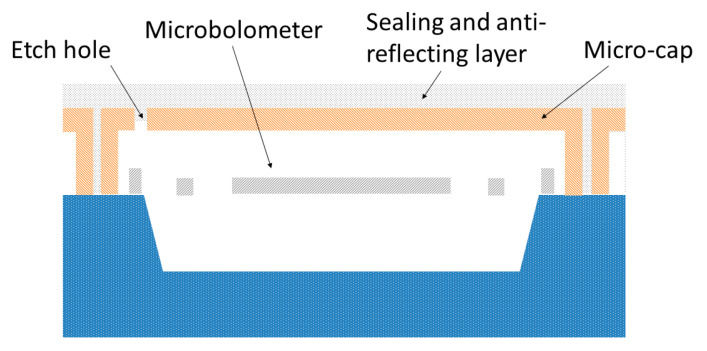
Schematics of the pixel level packaging technology.

**Table 1 micromachines-11-00800-t001:** Resistivity and temperature coefficient of resistance (TCR) of several complementary metal oxide semiconductor (CMOS)-compatible materials [12,75].

Material	Resistivity (10^−4^ Ω∙cm, at 300 K)	TCR (%/K)
undoped polysilicon	199	−0.085
n−polysilicon	62.2	−0.016
Al	0.03	0.38
Ti	1.2	0.25

**Table 2 micromachines-11-00800-t002:** Parameters of capacitive transimpedance amplifier (CTIA) read-out circuit in microbolometer infrared focal plane arrays (IRFPAs).

Reference	Analog Output Swing (V)	Power Dissipation (mW)	Integration Capacitance (pF)	Linearity	Supply Voltage (V)
METU [87]	2.5	85	1–32 (programmable)		3.3
ULIS [88]	2.8	150		≈1%	5
ULIS [89]	2.8	150		<1%	5
AUT [90]		351			3.3
WPU [91]	2.7	29.8	10		5

**Table 3 micromachines-11-00800-t003:** Parameters of a 14 bits on-chip pipeline analog-to-digital convertor (ADC) designed for microbolometer IRFPA [95].

Effective Number of Bits	Signal-to-Noise Ratio	Differential Nonlinearity	Integral Nonlinearity	Total Harmonic Distortion	Power Consumption
12.9	79 dB	0.76	4.9	−72 dB	95 mW

**Table 4 micromachines-11-00800-t004:** CMOS-compatible microbolometer IRFPAs versus other microbolometer IRFPAs.

Reference	Material	Array Size	Pixel Size	ROIC Type	Frame Rate ^1^	NETD
Mitsubishi [20]	Diode	320 × 240	17 μm	CTIA	60 Hz	50 mK
MikroSens [45]	CMOS-compatible resistance	120 × 160	35 μm	CTIA	11 Hz	117 mK
Toshiba [37]	Diode	320 × 240	22 μm	GMI	25 ms	200 mK
Raytheon [60]	VO_x_	640 × 512	20 μm		30 Hz	<50 mK
DRS [61]	VO_x_	1024 × 768	17 μm	DI	30 Hz	<50 mK
FLIR [99]	VO_x_	640 × 512	12 μm		60 Hz	<40 mK
L-3 Communications [85]	a-Si/a-SiGe	1024 × 768	17 μm	DI	10 ms	35 mK
ULIS [65]	a-Si	1024 × 768	17 μm	CTIA	30 Hz	46 mK

^1^ Or time constant in case the frame rate is not mentioned.

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
