# Peer review of "Low-Cost Microbolometer Type Infrared Detectors"

_micromachines, 2020, doi:10.3390/mi11090800_

Round 1

Reviewer 1 Report

Line 100...: The NETD formula should be checked again, e.g. not 4F but 4F+1, with F= 1 this is important. In the denominator the integral over the wavelength range (dL/dT) is missing.

Line 134…: The reduction of the pixel size has reached the physically given spatial limit. Smaller pixels no longer increase the spatial resolution of the image. (see your point 7.).

Figure 3 and 4: More contrast in the drawing.

Line 391: VGA format is 640x480. 1024x768 is XGA format.

Reviewer 2 Report

The manuscript presents a solid review of the state of the art in low-cost CMOS-compatible microbolometers. Overall, in my opinion it will be useful to many readers and is likely to attract a fair number of citations. I do not find any glaring omission in the presentation of the specific sensor technology chosen by the authors for their review. I find nonetheless some specific aspects of the presentation that may be improved and, thus, I provide below my corresponding suggestions. Besides (and although the language usage is usually clean and easy to understand) some grammatical or spelling issues surface here and there in the text. Because of that I recommend revision of the manuscript by a native English speaker, or a professional translator.

- The paper mentions photon IR detectors as an alternative sensor, mainly in the introduction, but it does not define them. I think a short sentence is needed to provide such definition, even if only a first approximation one, and also a reference.

- The paper does not mention the existence of superconducting (low- or high-temperature) thermal bolometers, not very dissimilar in working principle to the CMOS ones reviewed here but with markedly different specification numbers (importantly including the need to be cryocooled). They should mention that alternative bolometer, and provide a reference.

- I find that the English usage is particularly problematic in the following lines, that I recommend to re-writte even before they are tested by some native English speaker: lines 39, 43, 79, 173, 227, 328.

- The authors should test if they defined all the acronyms they use. I think MEMS is undefined, for instance.

- Authors should clarify in lines 102 and 103 what they mean by "target". Also its temperature (is it the actual one or the luminous equivalent as per the black-body radiation?)

- In Figures 1 and 2, authors should include (in their captions) citations to the references from which they obtain the data. They must also indicate what data is obtained from what reference (for instance as follows: "data taken from (in left-to-right order) [ref1], [ref2], ..."). In Fig. 2 I find redundant the "NETD" inset, that could be supressed.

- In the Figure 3, the different colors should be given numbers, in addition to their current lables (example: "(1) Si substrate"). Then, in the first paragraph of section 3.1, the one that explains that figure, these numbers should accompany each text descriptions, to make it easier for the reader.

- The same applies to Figure 4 and the descriptions in lines 154 to 170

- Figures 3 to 5 would greatly benefit if a small inset could be added with a 3D perspective drawing showing a top and lateral view (i.e.: a view as in Fig. 6)

- In Figure 6, I understand that "matel pattern" should be "metal pattern"

- In line 298 (sect. 4.2), I think "Sigma-Delta" is undefined in the manuscript.
